# Communication about Prognosis and End-of-Life in Heart Failure Care and Experiences Using a Heart Failure Question Prompt List

**DOI:** 10.3390/ijerph19084841

**Published:** 2022-04-15

**Authors:** Lisa Hjelmfors, Jan Mårtensson, Anna Strömberg, Anna Sandgren, Maria Friedrichsen, Tiny Jaarsma

**Affiliations:** 1Department of Health Medicine and Caring Sciences, Linköping University, 581 83 Linköping, Sweden; lisa.hjelmfors@liu.se (L.H.); anna.stromberg@liu.se (A.S.); 2Department of Nursing, School of Health and Welfare, Jönköping University, 551 11 Jönköping, Sweden; jan.martensson@ju.se; 3Department of Cardiology, Linköping University, 581 83 Linköping, Sweden; 4Center for Collaborative Palliative Care, Department of Health and Caring Sciences, Linnaeus University, 351 95 Växjö, Sweden; anna.sandgren@lnu.se; 5Palliative Education & Research Centre, Vrinnevi Hospital, 601 82 Norrköping, Sweden; maria.friedrichsen@regionostergotland.se; 6Department of Advanced Palliative Home Care, Vrinnevi Hospital, 601 82 Norrköping, Sweden

**Keywords:** palliative care, question prompt list, heart failure

## Abstract

Background: To further advance the use of a heart-failure-specific question prompt list (HF-QPL) for communication about prognosis and end-of-life care, knowledge about such communication and the perceptions and experiences of professionals is needed. Objectives: 1. to describe health care professionals’ perceptions of communication about prognosis and end-of-life in heart failure (HF) care, and 2. to describe their experiences of using a HF-QPL. Design: A qualitative design that analyzed material from written assignments of nurses and physicians who were using a HF-QPL while participating in a communication course. Methods: Fifteen health care professionals from different regions in the south of Sweden were included. The data were collected from course assignments on 1. their reflection on the suitable timepoint for talking about prognosis for the first time, 2. their reflection on the HF-QPL, and 3. their experiences of using the HF-QPL in clinical practice. Data were analyzed using thematic analysis. Results: Five overarching themes were identified. The first theme was awareness of professional role responsibilities that described the recognition of different responsibilities in these conversations within the HF team. The second theme described the importance of being optimally prepared, and the third that confidence and skills are required to use the HF-QPL. The fourth theme described the HF-QPL as a bridge in the communication between professionals, patients, and family members. The fifth theme identified challenges using the HF-QPL in HF care. Conclusions: Using a HF-QPL in HF care has the potential to start conversation and facilitate discussion about the HF trajectory.

## 1. Introduction

Patients with chronic heart failure (HF) have been described as having little insight into the trajectory of their illness and lacking information on their prognosis [1,2]. Many patients might not even realize the seriousness of their condition, which could have an impact on how they cope with their illness and make plans for the near future. Although not all patients want to think or talk about the HF trajectory, others would welcome conversations about prognosis and end-of-life care [3,4,5].

However, prognosis and end-of-life care are not routinely discussed in HF care, often because health care professionals struggle with how to initiate these conversations without taking away the hope from the patients and their families [6,7]. Reluctance to discuss prognosis and end-of-life care is also connected with health care professionals’ attitudes and views of palliative care and their own fears of death and dying [8]. Furthermore, health care professionals have reported a need for support and knowledge of how to discuss these often sensitive topics and stated that they are easier to discuss if the patient and the family themselves take the initiative to start a discussion [9].

There are several approaches to support patients and families to be active in discussions and involved in the care. One successful approach in cancer care is the use of a question prompt list (QPL) [10], which is a list of questions that is provided to the patient and family members to identify questions they wish to ask the health care professional. Studies involving QPLs in cancer care report that patients and family members find such a tool useful to frame questions and prepare them for clinical visits.

The use of a QPL is not as evident in care for patients with HF. Since some issues might be specific to this patient group (e.g., symptoms, pacemakers, resuscitation), we developed a HF-specific QPL and a short course for health care professionals to optimize conversations about the HF trajectory [2]. To help future implementation of the HF-QPL and understand possible challenges, it is important to know the attitudes and perceptions of professionals about communication on prognosis and end-of-life care and learn from experiences of using such a HF-QPL. This knowledge can help us understand the need for specific training, support, or other interventions in broader implementation.

Therefore, the aim of this paper was two-fold: 1. to describe health care professionals’ perceptions of communication about prognosis and end-of-life in HF care, and 2. to describe their experiences of using a HF-QPL which aims to improve such communication.

## 2. Materials and Methods

### 2.1. Design

An inductive and descriptive qualitative design was used that analyzed material from written assignments of health care professionals (nurses and physicians) taking a communication course and using a HF-QPL.

### 2.2. Setting: Communication Course and the HF-QPL

The communication course for health care professionals had three learning objectives: to expand their knowledge about the HF trajectory and end-of-life care, to improve their knowledge, confidence, and skills in communication, and to provide them with a practical communication tool (the HF-QPL). The communication course was based on adult learning principles, using directed learning [11] with online lectures, homework, reflections, and reading assignments. The course also included one training day to practice discussing the HF trajectory and end-of-life care with simulated patients and family members (actors). Course leaders included a behavioral scientist and nurses and physicians with a background in HF and/or palliative care.

The HF-QPL used was an A4 booklet in a paper format containing 45 questions grouped into topics including: (1). HF and what to expect in the future, (2). help and support during deterioration, (3). end-of-life care issues, (4). questions for the family members, and (5). questions about pacemakers or defibrillators 2.

### 2.3. Sample and Procedure

An invitation letter was sent out to health care professionals in HF clinics who were previously recruited in a survey study in all regions in Sweden [7]. All professionals in cardiac care who were interested in the content of the course and the HF-QPL were eligible to participate in the course, and they could inform the research group of their interest. The course took place between 9 January until 26 February 2017 and was free of costs. In total, 21 professionals from different regions in the south of Sweden signed up for the course; six did not attend the course due to private matters or changes in their job situation or workload. Demographic data on the 15 course participants were collected via the online course platform before the beginning of the course. All participants were women and there were 13 nurses and 2 physicians, all with a long experience in HF care (Table 1).

### 2.4. Procedure and Data Collection

Course participants were first asked if their data could be used for research purposes. The written text from the individual assignments of the 15 participants was used as data material. Participants wrote their assignments (see Box 1) as a Word file and sent it via email to the course team. We also analyzed data from the course evaluation provided at the end of the course.

Box 1Description of assignments in the course.
**Assignment 1**: Based on your profession, think about the timepoint that you find suitable for talking about prognosis with a HF patient **for the first time**. Explain why you feel that the chosen timepoint is suitable.**Assignment 2**: Read through the QPL and think about what questions you have knowledge about and are able to discuss with the patient and their family members. Based on your profession, reflect upon what your role should be in the communication about prognosis and end-of-life. Select three questions from the QPL that you feel would be difficult to discuss with a patient and their family members.**Assignment 3**: Provide the QPL to one patient and/or one family members and let them have some time read it and highlight questions that they want to discuss. Discuss the selected questions. When the conversation has ended, write down what worked and what did not work during the conversation. Reflect upon your role in the conversation, your strengths and what you can improve upon. You are free to choose the timepoint you find the most suitable to give the QPL to the patient/family members.


### 2.5. Analysis

The qualitative data were merged and analyzed as a whole using thematic analysis [12]. Codes and themes were inductively developed from the data with the aim to identify and interpret patterns. Data were analyzed in a five-phase process (Figure 1). The first phase included familiarizing ourselves with the data. In the second phase, extracts of data were labeled with a code that captured semantic or latent meanings. Then, codes were analyzed and collated into overarching themes that captured the central organizing concept, not separating the two aims. Then, the overarching themes were further reviewed in relation to the coded data and the entire data set. Coding and development of overarching themes was primarily conducted by LH, JM, and TJ. In the last step in which all co-authors participated, the final analysis was reviewed and discussed until consensus was reached to enhance trustworthiness.

### 2.6. Ethical Considerations

The study was approved by the Regional Ethical Board in Linköping (Dnr. 2013/244–31). The participants were informed about the confidential handling of study data, and informed consent was obtained from all participants, with them agreeing that the information provided could be used for research purposes.

## 3. Results

Five overarching themes were identified. The first theme was awareness of professional role responsibilities, which described the recognition of different responsibilities in these conversations within the HF team. The second theme described the importance of being optimally prepared, and the third that confidence and skills are required to use the HF-QPL. The fourth theme described the HF-QPL as a bridge in the communication between professionals and patients and family members. The fifth theme identified challenges using the HF-QPL in HF care.

### 3.1. Awareness of Professional Role Responsibilities

The professionals described that discussing prognosis and end-of-life care is an important part of their work but that they have different responsibilities in these conversations. Physicians were perceived to be responsible for providing information on the prognosis to the patient, and the nurses’ responsibility was to function as a support, to inform and explain, mainly after the first initial conversation had been held by the physician.
*Spontaneously, it feels as if the main responsibility for talking about prognosis is expected to lie with the physician. In breakpoint conversations in the ward where I work, we usually include both physicians and nurses, but at least in the first session it is mostly the physician who talks about prognosis. The nurse sits along as an extra ear and sounding board in that situation, both for the patient’s sake and for our own, for feedback and for questions about care.**(Physician)*

It was suggested that the nurse functions as a “spider in the web,” recognizing needs and wishes of the patient and family, as the nurse often spends more time with them than the physician. The professionals also suggested that the nurse could be the one who prepares a conversation about prognosis and end-of-life care, making sure that the physician takes the time to discuss the questions the patient and family might have. However, the professionals described that both the physician and the nurse have the obligation to provide honest information about the prognosis and end-of-life care to the patient and the family. Such conversations can be challenging since the professionals want the patient and the family to keep the hope for the future.
*As a nurse, you have an important role in informing and discussing with patients and their loved ones about prognosis and the end of life. I think all patients have the right to have one or more proper conversations with a physician who is familiar with the patient. But after that, we nurses have a great importance for patients and relatives. We see the patient more and usually get a little closer to the patient, get to know them.**(Nurse)*

### 3.2. The Importance of Being Optimally Prepared

The professionals described that a good conversation about prognosis and end-of-life care involves a professional who is prepared to discuss these difficult topics. The conversation should be held in accordance with the patient’s and the family’s preferences and wishes.
*I felt there was a need to discuss his prognosis and I had prepared well by thinking what I wanted to say and what questions he might ask.**(Nurse)*

The professionals also described that part of being prepared for the conversation included practical issues such as having enough time to talk, preferably in a private room without disturbance. They also thought that a conversation about prognosis and end-of-life care could take place early in the HF trajectory and even during the first contact, as this might help the patient and the family to prepare for the future. However, they also emphasized that the preferences of the patient and the family should always lead the conversation and the professional should adapt the information accordingly. Patients and family might want different information at different time points and that needs to be acknowledged and respected. A challenge in communication is to keep the balance between the obligation as a professional to provide honest information to the patient and the family, but at the same time respect how much information they want.
*It is so important not to impose any information that a patient is not prepared for. How do you keep that balance? I think that is by far the most difficult. Sometimes I feel inadequate and like I haven’t done my job in case I haven’t done my “duty” and told a patient that he/she is going to die. But just as often I think, when I (after giving a difficult message or having a conversation about progress of a serious chronic illness) have tried in several different ways to find out if the patient has more thoughts and still gets evasive answers, they probably are not fully prepared to hear a straight answer either but need to deal with the situation in a different way. And it’s my duty to feel that need. I’m not sure what’s right or wrong. Is it always right to wait? Is it always wrong to rush it? The patient dies without knowing (or does he/she really?) Is this always bad? **(Physician)*

The family can be encouraged to participate in the discussion if the patient wishes. If the professional has a good relationship with the patient and the family, the participants perceive that there is a greater chance for a good conversation where the patient and the family feel acknowledged, supported, and safe.

### 3.3. Confidence and Skills Are Required to Use the HF-QPL

The professionals described that they felt ready to discuss questions in the HF-QPL when they had enough confidence and skills. To give out the HF-QPL to patients and family, confidence was needed to discuss any question that might come up without being afraid of how to answer.

Some professionals also described that it was possible to feel confident in conversations even if they could not always answer all questions; they thought it was fine not knowing all the answers.
*I felt it became an honest communication. My experience allowed me to give a straight answer to the questions without feeling stressed. Some questions don’t have an easy answer, but it felt like the patient understood that.**(Nurse)*

Topics concerning HF and its impact on daily life and on help, support, and treatment were most often reported as topics that the professionals felt confident and skilled to discuss with patients and families.

To be ready to use the HF-QPL, the professionals also described that they needed skills to know what to communicate and how they could communicate with the patient and the family in a good way. The skills to communicate included, for example, being empathic to patients and families, to center the communication around their emotions, and always provide information at the patient’s level.
*One should be responsive and respect what and how much the individual patient wants to know. Some patients do not want to know anything at all, and you then must meet the patient where they are and talk about what they want to talk about.**(Nurse)*

The communication skills were also described by the professionals as having knowledge in motivational interviewing and knowledge on useful phrases to initiate a communication.

### 3.4. The HF-QPL Is a Bridge in the Communication

The professionals described how the HF-QPL was a guide in the conversation and how it seemed to help the patients and the family to pose questions that were important to them, which could lead to detailed discussions. Professionals doubted when it was the optimal time to give the HF-QPL to patients and families, but one option was when the professional thought it was needed, depending on each specific patient. For example, before a “breakpoint” conversation or at repeated exacerbations.

They described how the patient and the family chose which questions to discuss and that they took the lead in the conversations and received new information.
*It felt good to have the tool included as support. It became like a bridge between us, and we could browse it together and think and talk about it. I felt the conversation was getting less charged because of it. Without the tool, it would have been difficult to come up with questions to ask, I think, and it would probably have been difficult for the patient and relatives to get started talking as well.**(Nurse)*

### 3.5. Challenges of the Question Prompt List

The professionals described that it was difficult to use the HF-QPL if they did not feel ready or were even afraid of what questions or emotions might come up in a conversation. If they lacked knowledge, it was difficult to have a discussion and feel comfortable and secure, and if the professional could not answer questions, it felt insufficient. Many professionals reported a lack of knowledge of how to discuss questions about devices, which made it hard to discuss these issues.
*Can we really know when a patient is going to die? How does the patient react when the pacemaker is switched off? How long will it take? We are usually unable to answer these questions. We leave the patient without them getting an answer, which makes things worse. I want to be able to give answers, be able to comfort, but how do you do it when you don’t know? **(Nurse)*

The professionals also described that questions that concerned the end-of-life care could be difficult to discuss, as these questions often do not have a simple answer if there was an answer to give at all.
*Questions about the last days of life, I think, are more difficult, partly because these are questions that feel difficult/tough and partly because these are topics that I am not used to talking about; these are also questions that may not have any real answers.**(Nurse)*

Some professionals experienced that they would prefer to have separate conversations with the patient and the family member(s), as they may have different information needs and preferences in the communication. Wishes from the patient to be cared for at home might be hard to discuss if the family member did not have the ability to take care of the patient and felt guilty about it.

Another experience was that some family members took over the conversation and left the patient in the background, making it difficult to focus on the patient in the conversation.

Some professionals were also concerned about giving out the HF-QPL to patients who seemed “too healthy,” or where in the beginning of their HF trajectory.

Another fear expressed by the professionals was how the patients and families would react to the HF-QPL and its content.
*The first thought when reading the brochure is Oops, before this is provided, the patient and their relatives must at least have received some information about the diagnosis, otherwise they will probably get a shock! **(Physician)*

## 4. Discussion

This paper provided further insight into the challenges health care professionals experience in discussing the HF trajectory, including prognosis and end-of-life in HF care. We also found that using a communication tool such as a QPL was perceived by health care professionals as promising and useful, which also was supported by our previous research [2,13]. A recent publication described the experiences with a pre-visit QPL to determine the educational needs of patients with HF and their family members. They found that one of the most selected topics from the prompt list was conversations about prognosis [14].

Regarding our first theme being the awareness of professional role responsibilities, our data confirmed that the perception of the responsibility of a professional in discussing prognosis and end-of-life care is important to consider. The composition of HF teams varies, as do the expectations of each team member and the education and experience in a team. Our study took place in a Swedish context where roles and expectations might be different than other countries. Nurses and doctors were described as having different responsibilities in conversations about prognosis and end-of-life care; however, they should work closely together. This underlines the importance of multi-professional teamwork where professionals have different skills and educational background that determines their roles and responsibilities [14,15,16]. We previously described in another sample of Swedish and Dutch HF nurses that some nurses perceived these discussions to be beyond their responsibility and authority and that physicians should be the first to discuss prognosis and end-of-life care with the patient [9]. Instead of applying dogmatic rules, it is recommended to start with open discussion in an HF team on expectations within a team over ‘who discusses what’ [5]. These policies can be different depending on culture [17] and country, skills, and education in the team as well as the kind of relationship the HF team members have with patients and families.

We also confirmed that a good conversation about the HF trajectory with patients and their family members requires ‘investment’ on a professional and organizational level. The second theme described the importance of being optimally prepared, and the third that confidence and skills are required to use the HF-QPL. On a professional level, knowledge, confidence, and skills are needed. On an organization level, it is necessary to provide professionals with time, room, and information material for patients and families. Sometimes, HF-specific patient education materials can be used. For example, the website heartfailurematters.org has specific sections such as “planning for the end of life” and “palliative care,” that professionals can refer patients to as a first step to gather information about these issues [18]. Another tool that can be helpful is the HF-QPL, as discussed in this paper. The HF-QPL was perceived as useful to identify knowledge gaps, lack of confidence, and skill in the professionals in discussing the HF trajectory. The HF-QPL also helped to keep the focus in the conversations and was helpful for important questions in the team that patients and family might want to discuss, as found in the fourth theme that described HF-QPL as a bridge in the communication between professionals and patients and family members.

The timing of conversation about prognosis and introducing a palliative care approach was a relevant issue in our data. As course participants reflected, there is not one optimal time for all patients to discuss, and even per patients, several discussions might be needed. ‘Upstreaming’ palliative care has been discussed in guidelines and statements, but there still seems to be reluctance to introduce such conversations early in the HF trajectory [5,19,20,21]. In a previous study, when cardiologists were asked about timing of discussing prognosis, almost half of them (47%) stated that prognosis should be discussed at the time of diagnosis, although 18% found the first period of decompensated HF or HF hospitalization as the best time to discuss it with the patient. Sixteen percent wanted to discuss the prognosis only in case of a serious decline in the HF condition [16]. However, if a HF-QPL is provided, the decision whether to discuss prognosis and end-of-life care lies with the patient and the family members, as they are the ones asking the questions. 

A fifth theme identified challenges using the HF-QPL and calls for careful implementation strategies and further research. In a recent letter to the editor of Journal of Palliative Medicine, McDarby et al. proposed a research agenda for the QPL in outpatient palliative care with future aims and research questions [22]. The authors of the research agenda stated that there is still a knowledge gap regarding determining the ideal timing and delivery of QPLs, as well as over the patient and family member populations most likely to utilize and benefit from a QPL. Previous studies on QPLs from oncology described distributing the QPL to the patient and family members 1–2 weeks [23] up to 1 month before the clinical visit to give them more time to read and think about the possible sensitive topics [24].

### Methodological Reflections

The course took place in 2017 and the data were analyzed in 2021. In the 4 years between data collection and analysis, the researchers had other responsibilities in their daily work. However, since we noticed that HF-QPLs are still not routinely used in HF care, we considered the data still relevant and believe that our results can be used to improve current care.

There were no specific entry criteria to the course, and we managed to recruit 15 participants to the course. Since this was the first time we ran the course, we considered these 15 participants sufficient to get feedback on the overall satisfaction and feasibility of the course. This limited number and the fact that we only had two physicians can limit transferability of the results and needs to be considered as potential limitations.

We also only had female participants in the course with a long experience in care, which might reflect current practice in HF specialist care but also limits transferability. Nevertheless, the data that were obtained from the course participants were rich in content. Another limitation is that we used written data from assignment and evaluation forms, which did not give us the opportunity to ask follow-up questions. On the other hand, it can be seen as positive that using this material might reflect ‘real practice’ better than recruiting professionals specifically for a study. However, interviews probably would have given richer data and at least would have allowed the opportunity to ask more follow-up questions.

## 5. Conclusions

Combining our previous research on the HF-QPL with the current findings, we concluded that using a HF-QPL for patients with HF and their families has potential to start conversation about the HF trajectory, to function as a bridge between patient and health care professional and facilitate a conversation where the patient and the family feel acknowledged, supported, and safe.

## Figures and Tables

**Figure 1 ijerph-19-04841-f001:**
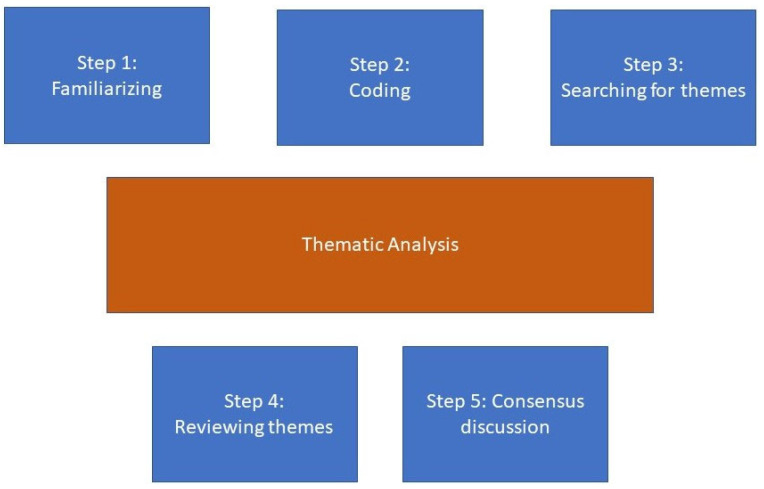
Stages of coding and analysis of the data.

**Table 1 ijerph-19-04841-t001:** Demographic data of the study participants.

Female sex, n (%)	15 (100%)
Age, mean ± sd	40 ± 12
Occupation, n (%)	
Nurse	13 (87%)
Physician	2 (13%)
Workplace	
Hospital	13 (87%)
Primary care center	2 (13%)
Years working in health care, mean ± sd	16 ± 13
Years working with patients with HF, mean ± sd	9 ± 8
Work percentage working with patients with HF per week (range)	5–100%

sd standard deviation, HF heart failure.

## Data Availability

The participants were informed about the confidential handling of study data, and informed consent was obtained from all participants, with them agreeing that the information provided could be used for research purposes.

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
