# Peer review of "Communication about Prognosis and End-of-Life in Heart Failure Care and Experiences Using a Heart Failure Question Prompt List"

_ijerph, 2022, doi:10.3390/ijerph19084841_

Round 1
Reviewer 1 Report
This manuscript is a paper on the presentation and evaluation of a question prompt list (HF-QTL) on heart failure for nursing and care professionals. HF-QTL is considered to be an effective method for early detection of disease signs to formulate the question to some extent through discussions between field staff and experts.
The authors constructed HF-QTL by referring to cases of other diseases (cancer), and is proceeding with improvement through discussions with field experts. It is clear from the references that the setting of such question items is important in the fields of home care and medicine.
However, in order to present HF-QTL as a research paper, discussions based on objective data are required. As an objective argument, one of the following is considered appropriate.
1 Effectiveness of the HF-QTL analysis are shown statistically aggregating and analyzing the opinions of field experts on disease signs.
2 Analyze duplication of the question-points (including disease signs) that require prediction of Heart Falure in the QTL. The question items are analysed using language processing and semantics, and construct the most efficient and question items on the least number of questions.
3 After collecting patient data by HF-QTL, the relationship between the answers to each question and the patient's outcome are analysed using correlation and machine learning.
Unfortunately, this manuscript does not include such a quantitative analysis.
Therefore, this manuscript meets the criteria required by this journal.
Please add the objective analysis required as a research paper and resubmit the research.
Reviewer 2 Report
Summary:
In the present paper, the authors present their research results that was carried out in order to identified the shortcomings of communications with final stage HF patients and their family members.
The theme is very interesting and current, the study is original and the study design is appropriate. The methods are clearly described and the manuscript is well structured. The conclusions of the work are clear and are well supported by the results. The authors contribute to this field of research with new and valuable findings. Nevertheless, some revisions are recommended before publication.
Observations:
Line 91: “All professionals within cardiac care” could be changed to “All professionals in cardiac care.”
Line 92: “...and they could send their interest to the research group.” could be changed to “...and they could inform the research group of their interest.”
Line 99: The period after “(Table 1)” isn't absolutely necessary. The references are also written after the end of sentences, so here you could follow the same formula.
In Line 88, 112 and 113 the format of the references is not consistent with the rest of the paper where the numbers of the references are written within square brackets. If these numbers within round brackets are not references, please clarify what they are supposed to mean.
Line 224: “choose” should be “chose”.
Line 322: “then” should be “than”.
Table 1: In the first row you start with a clearly followable format regarding the understanding of the data. You wrote “Female, sex, n (%)” and the numbers follow accordingly. But then the rest of the table doesn't follow this clear format. I suggest you fix the Table 1 as follows:
Female, sex, n (%)
Age, mean ± sd
Occupation, n (%)
Workplace, n (%)
Years working in health care, mean ± sd (here the numbers should also be fixed into 16 ± 13)
Years working with patients with HF, mean ± sd (here the numbers should also be fixed into 9 ± 8)
Reviewer 3 Report
The manuscript describes the perception and experiences of healthcare professionals regarding communicating prognosis and end-of-life heart failure care, as well as using a question prompt list. It followed a descriptive qualitative design based on thematic analysis. The manuscript is well written and brief. The topic is quite interesting as well. However, it presents some methodological issues and certain weaknesses when presenting results and discussions which make it unsuitable for publication in its current form. The following are some observations.
According to the authors, the data were obtained in 2017. In this regard, what is the practical and methodological justification for a long period of time between obtaining the data and submitting the manuscript?
Authors obtained a cross-sectional sample of 15 healthcare professionals. There is not a specific procedure to determine a sample size for thematic analysis. Some literature indicates a 10-50 sample size for participant-generated text; however, Fugard and Potts (2015) presents different sample sizes as a function of the theme prevalence and how many instances of the theme are desired.
In this regard, the authors stated “Five overarching themes were identified”. Thus, what is the rationale for a 15 participants sample size? The answer for this question is related to the time period elapsed since the data were obtained. Is there any chance that the results might be different if the study would have been done recently? If so, during the 5-year period since the sampling, why did the authors not increase the sample size to obtain more data? What is the rationale for not conducting a longitudinal study?
A graphical representation of the method followed might clarity the manuscript.
The discussion section presents few citations. It did not sufficiently contrast the manuscript’s results with previous or similar approaches. In fact, the paper is lacking literature on the subject from different approaches. The paper is based on only 18 references, which for the topic’s importance seems to be an inadequate number. Particularly, it did not refer to previous studies from the IJERPH.
Authors concluded “This study showed that using a HF-QPL for patients with HF and their families has potential to start conversation about the HF trajectory, to function as a bridge between patient and health care professional, and facilitate a good conversation where the patient and the family feel acknowledged, supported, and safe.”
Given the nature of the study, the manuscript did not present any form of validation whether using HF-QPL indeed facilitates such potential benefits mentioned in the conclusion, nor did authors propose future ways to validate those benefits. Thus, the implication of the manuscript seems limited.
Communication deficiencies
The manuscript utilizes reference numbers placed in square brackets (Required by IJERPH). However, the manuscript also presents parenthesis, which may be typos. The following are some examples.
Page 2, row 88 “questions about pacemakers or defibrillators (2).”
Page 4, row 112 “(12) Codes and themes were inductively developed from the data,”
Page 4, row 113 “Data was analyzed in a five-phase process. (12)”
Andrew Fugard and Henry Potts (2015) Supporting thinking on sample sizes for thematic analyses: a quantitative tool. International Journal of Social Research Methodology, 18:6, 669-684. DOI: 10.1080/13645579.2015.1005453
Round 2
Reviewer 1 Report
First of all, I would like to apologize to the authors. As a Reviewer, I reviewed the manuscript first and gave a low rating without knowing how to evaluate Qualitative research at all.
After I read the author's comment on my first-round comment, on which the anthers emphasized qualitative methodology, I considered the requirement for understanding qualitative methodology in IJERPH. So I read qualitative-studies published in IJERPH, and considered about the purpose and evaluation criteria in qualitative methodology.
As a result, I understand that this manuscript described the perspectives that physician and nurse should know. These narratives and conversation were corrected through the question HF-QPL. On the perspective I understood that the issues were organized into important focuses on physisian and nurse. The conversations started from HF-QPL were organized by theme. The professionals on HF would understand important themes through conversations and narratives present in Results. Description were well organized for such professionals.
Although this paper is not quantitative and does not show a simple and generalizable laws, but it would be understandable and meaningful to professionals, patients and their families. Thus descriptions in this manuscript is worth sharing to experts who face life-threatening illnesses. The theme and conversation are well constructed. Therefore, I recommend the acceptance of this manuscript for publish in IJERPH.
First of all, I would like to apologize to the authors. As a Reviewer, I reviewed the manuscript first and gave a low rating without knowing how to evaluate Qualitative research at all.
After I read the author's comment on my first-round comment, on which the anthers emphasized qualitative methodology, I considered the requirement for understanding qualitative methodology in IJERPH. So I read qualitative-studies published in IJERPH, and considered about the purpose and evaluation criteria in qualitative methodology.
As a result, I understand that this manuscript described the perspectives that physician and nurse should know. These narratives and conversation were corrected through the question HF-QPL. On the perspective I understood that the issues were organized into important focuses on physisian and nurse. The conversations started from HF-QPL were organized by theme. The professionals on HF would understand important themes through conversations and narratives present in Results. Description were well organized for such professionals.
Although this paper is not quantitative and does not show a simple and generalizable laws, but it would be understandable and meaningful to professionals, patients and their families. Thus descriptions in this manuscript is worth sharing to experts who face life-threatening illnesses. The theme and conversation are well constructed. Therefore, I recommend the acceptance of this manuscript for publish in IJERPH.
Author Response
Thank you for reviewing the paper with 'new eyes'. We really appreciate that, thank you for your feedback
Reviewer 3 Report
Thank you for the opportunity to review the edited version of the manuscript. First of all, I congratulate the authors for having improved their manuscript. Most of the observations were addressed. For example, the authors integrated more literature on the topic, which allow readers to compare the findings with previous studies. In addition, the authors now include a figure that depicts the followed method. However, there are some clarifications that might be included in the manuscript.
In the first place, the authors did not include explanations regarding the time elapsed since the data was collected and the practical and methodological justification for this long period in the manuscript. Readers might be interested in the rationale of this elapsed time. The exclusion of this information might be considered a limitation.
Second, the manuscript also lacks an explanation or a stated limitation from having a sample size of 15 participants.
By correcting these two observations, the manuscript would be adequate for publishing.
